# In Silico Screening of Metal−Organic Frameworks and Zeolites for He/N_2_ Separation

**DOI:** 10.3390/molecules28010020

**Published:** 2022-12-20

**Authors:** Ivan V. Grenev, Vladimir Yu. Gavrilov

**Affiliations:** 1Department of Physics, Novosibirsk State University, Pirogova Str. 1, Novosibirsk 630090, Russia; 2Boreskov Institute of Catalysis, Ac. Lavrentiev Av. 5, Novosibirsk 630090, Russia

**Keywords:** helium recovery, adsorption-based separation, membrane-based separation, metal−organic frameworks, zeolites, in silico screening, molecular simulation

## Abstract

In silico screening of 10,143 metal−organic frameworks (MOFs) and 218 all-silica zeolites for adsorption-based and membrane-based He and N_2_ separation was performed. As a result of geometry-based prescreening, structures having zero accessible surface area (ASA) and pore limiting diameter (PLD) less than 3.75 Å were eliminated. So, both gases can be adsorbed and pass-through MOF and zeolite pores. The Grand canonical Monte Carlo (GCMC) and equilibrium molecular dynamics (EMD) methods were used to estimate the Henry’s constants and self-diffusion coefficients at infinite dilution conditions, as well as the adsorption capacity of an equimolar mixture of helium and nitrogen at various pressures. Based on the obtained results, adsorption, diffusion and membrane selectivities as well as membrane permeabilities were calculated. The separation potential of zeolites and MOFs was evaluated in the vacuum and pressure swing adsorption processes. In the case of membrane-based separation, we focused on the screening of nitrogen-selective membranes. MOFs were demonstrated to be more efficient than zeolites for both adsorption-based and membrane-based separation. The analysis of structure–performance relationships for using these materials for adsorption-based and membrane-based separation of He and N_2_ made it possible to determine the ranges of structural parameters, such as pore-limiting diameter, largest cavity diameter, surface area, porosity, accessible surface area and pore volume corresponding to the most promising MOFs for each separation model discussed in this study. The top 10 most promising MOFs were determined for membrane-based, vacuum swing adsorption and pressure swing adsorption separation methods. The effect of the electrostatic interaction between the quadrupole moment of nitrogen molecules and MOF atoms on the main adsorption and diffusion characteristics was studied. The obtained results can be used as a guide for selection of frameworks for He/N_2_ separation.

## 1. Introduction

Helium is widely used in aviation, space and electronic industries, science and medicine due to its unique physical properties [1,2,3]. Today helium is mostly recovered from natural gas by the cryogenic distillation method. The traditional process used for helium separation [3,4,5] consists of the following stages: preliminary purification (separation from C_2+_ hydrocarbons, water, H_2_S and CO_2_), cryogenic separation of methane followed by cryogenic separation of nitrogen in a nitrogen rejection unit (NRU) yielding feedstock with helium concentration 1–3%, separation of the N_2_/He mixture in a helium recovery unit yielding 50–70% crude helium vapor stream, purification of crude helium in a helium upgrade unit producing outcoming gas with helium concentration about 90%, deep purification in a helium purification unit producing helium with 99.995% purity and liquefaction of commodity helium. Membrane-based separation, vacuum swing adsorption (VSA) and pressure swing adsorption (PSA) processes are alternatives for the cryogenic distillation method [3,6,7]. Specific feature of these methods is that they can be used both as independent technological processes for helium recovery directly from the natural gas or as intermediate stages at existing factories using cryogenic distillation. According to the existing estimates [8,9], adsorption-based and membrane-based technologies are most expedient to be used at the stages of crude helium separation from the N_2_/He mixture and deep purification. The efficiency of these processes is mostly determined by the physical properties of sorbents and materials used to produce membranes. In the case of the swing adsorption separation method, the sorbent must have high adsorption selectivity and working capacity. In the case of membrane-based separation, the material should possess high membrane selectivity and permeability [10,11,12,13]. Therefore, the search and development of novel efficient materials for both membrane-based and adsorption-based separation of the N_2_/He mixture is an important problem.

Due to the diversity of the framework types, varying cation composition, high adsorption capacity, thermal stability and relatively low cost, zeolites are widely used in industry as sorbents in adsorption-based gas separation processes. Commercial zeolites potentially applicable for N_2_/He separation are ZSM-5, HISIV 3000 (UOP), 5A (UOP, Sigma, St. Louis, MO, USA) and 13X (UOP) [3]. It is very difficult to estimate the adsorption selectivity of porous materials in separation of He and N_2_ under ambient conditions because helium is poorly adsorbed on them. We are aware of only one paper [14] reporting an experimental estimation of helium sorption on zeolites 13X, 5A and 4A at room temperature and atmospheric pressure. Using these data as well data on nitrogen adsorption on these materials [15], it is possible to estimate their adsorption selectivity as 221, 458 and 54 for zeolites 13X, 5A and 4A (Appendix A), respectively. For membrane-based separation, zeolites with framework types of DDR, MFI, STT and CHA are used. High membrane selectivity to helium can be achieved using porous frameworks with effective pore diameter close to the kinetic diameter of the nitrogen molecule. Polycrystalline membranes always have defects in the crystallite packing resulting in the loss of selectivity and increase of permeance. For example, a membrane based on zeolite STT [16] having strong size-exclusion effect for nitrogen demonstrated the He/N_2_ selectivity equal to 11 at the average thickness of 2 μm and 59 at the average thickness of 6.6 μm. Zeolite 4A has weaker size-exclusion effect resulting in lower membrane He/N_2_ selectivity ranging from 1.4 to 3.7 [17,18,19,20]. Membranes from zeolite DDR have selectivity close to 3 [21,22]. At low temperatures, zeolites membranes are nitrogen selective, since adsorption selectivity dominates over diffusion selectivity. The dependence of the membrane selectivity on temperature for ultrathin MFI membranes was evaluated in a wide temperature range by Yu et al. [23]. It was shown that the N_2_/He membrane selectivity reaches 52 at 174 K for equimolar CH_4_/N_2_/He mixture at 3 bar feed pressure and 0.2 bar permeate pressure.

Metal−organic frameworks form a promising class of crystalline porous materials that can be used in the adsorption-based and membrane-based gas separation. The great variety of available organic and inorganic building blocks makes it possible to tune their adsorption properties, surface area, pore sizes and pore volume. Compared to zeolites, in the literature there are not as many experimental data on the selectivity and permeance of MOF membranes for the He/N_2_ separation. MOF that is most widely used for synthesis of membranes is ZIF-8. According to the literature data [24,25,26], polycrystalline membranes made of ZIF-8 with the thickness varying from 17.5 to 80 μm are characterized by the He/N_2_ selectivity in the range of 4.22–5.45 and He permeability between 3137 and 11354 Barrer. ZIF-8 single crystals with the size of 500 μm were used as membranes [27]. The He/N_2_ selectivity of two studied single-crystal membranes was equal to 74.5 and 77.7 with the He permeability of 1935 and 2309 Barrer. Note that ZIF-8 has channels with an aperture of 3.4 Å, which is smaller than the kinetic diameter of the nitrogen molecule. The N_2_ diffusion in the framework is possible due to the structural flexibility of ZIF-8. Hindered N_2_ diffusion is reflected in low N_2_ permeability equal to 26.0 and 29.7 Barrer for two different single-crystal membranes. Another studied MOF is HKUST-1. A polycrystalline membrane made of this material with the thickness of 40 μm was prepared on α-Al_2_O_3_ support [28]. Such membrane had the He/N_2_ selectivity of 3.7. In the literature, there are also data on the He/N_2_ selectivity of a membrane made of IRMOF-1, which was equal to 2.4 [29]. Another representative of the IRMOF family IRMOF-3 demonstrated the selectivity of 2.5 [29]. In addition, the He/N_2_ selectivity of membranes made of MIL-53 [30] and [Cu_2_(bza)_4_(pyz)]_n_ [31] was reported to be equal to 2.4 and 3.9, respectively.

If the experimental measurement of the selectivity, adsorption capacity and permeability is not possible, simulation of adsorption and diffusion characteristics can give valuable information for estimation of the suitability of different materials for the He/N_2_ separation. The development of zeolite database IZA [32] and MOF databases CSD MOF [33] and CoRE MOF [34] containing the unit cell parameters and atomic coordinates initiated in silico screening of efficient materials for adsorption-based [35,36,37,38,39] and membrane-based [40,41,42] separation. For example, Zarabadi-Poor et al. carried out in silico screening of almost 500 MOF structures from the DFT-optimized CoRE MOFs database [43] for helium separation from natural gas via N_2_/He separation [44]. The API (Adsorbent Performance Indicator) metric [45] and membrane selectivity were used to determine the top performing MOF structures in the case of adsorption-based and membrane-based separation, respectively. We used the results obtained in this work as a reference for comparison. Most of the studies are focused on the screening of materials for membrane-based helium recovery from natural gas [10,46]. Kadioglu et al. screened 139 MOF structures for He/CH_4_ membrane-based separation [47]. It was shown that the top performing MOF structures had a pore limiting diameter in the range of 3.8–4 Å. Qiao and coworkers predicted gas permeability and selectivity of MOFs for membrane-based separation of He/CH_4_ and He/N_2_ at infinite dilution condition [48]. Based on the obtained results, the Top 5 MOF structures for He/N_2_ separation show selectivity in the range of 3.02 to 3.53 and permeability in the range of 8.72 × 10^2^ to 8.72 × 10^3^ Barrer. Daglar et al. studied incorporation of MOF fillers into polymers to obtain mixed matrix membranes (MMM) for 11 different gas separations processes [49]. It was shown that all MOF fillers improve He permeability with slight changes in He/N_2_ membrane selectivity. Helium permeability and He/N_2_ selectivity of such MMM varied from 39.2 to 1.13 × 10^4^ Barrers and from 0.8 to 622, respectively. Gas permeabilities and selectivities for covalent organic frameworks (COFs) from the CURATED COF database [50] and COF/polymer MMMs for helium separation were predicted in two studies by different groups (Aydin et al. [51] and Feng et al. [52]). It was shown that COFs had a linear correlation between He permeability and He/N_2_ selectivity. For He/N_2_ separation, the selectivity of COF structures varies from 5.8×10^–2^ to 8.7. In both studies it was shown that the addition of COF fillers improved the He permeability of MMMs without significantly changing the He/N_2_ selectivity. Based on the results of these studies, we think that MOF/COF membranes have low He selectivity at room temperature, while polymer membranes with high He/N_2_ selectivity have low He permeability. Thus, two strategies are possible for creating efficient membranes for the He/N_2_ separation: incorporation of porous fillers into the polymers to obtain MMMs with high He/N_2_ selectivity and He permeability, or the use of nitrogen-selective porous medium membranes. Since the first approach has been extensively studied in the literature, we have focused on the screening of nitrogen-selective membranes. The goal of this study was to perform screening of zeolites and MOFs for search of relationships between their structural parameters and performance characteristics for adsorption-based and membrane-based He/N_2_ separation.

## 2. Results and Discussion

### 2.1. Adsorption-Based Gas Separation

To test the force field models used in this study, calculated N_2_ adsorption isotherms for a number of well-known MOFs at 295–298 K and pressure up to 20 bar were compared with the corresponding literature data [53,54,55,56,57,58,59]. Figure 1 (left) demonstrates that the suggested force field model can be used to predict the nitrogen adsorption in various MOFs with good precision. Several force fields were used to simulate the nitrogen adsorption in zeolites: Dreiding [60], TraPPE [61] and the force field developed by Vujić and Lyubartsev [62]. For each force field model, N_2_ uptake values at 303 K and 1 bar pressure were calculated for five different pure silica frameworks, and the obtained results were compared with the corresponding experimental data [63] (Appendix A). The model by Vujić and Lyubartsev was used hereafter because it predicts the experimental data with the best precision. Figure 1 (right) demonstrates that this force field makes it possible to predict nitrogen adsorption isotherms with good precision both on pure silica zeolites and on several aluminophosphates [64,65]. The experimental conditions used to measure the nitrogen adsorption isotherms and the corresponding references are reported in Appendix A. Testing the used force field models for helium adsorption simulation is very difficult because the value measured in the adsorption experiment is excess adsorption. However, before the experiment, dead space is measured using the same helium. So, it is not possible to measure the helium adsorption isotherm by traditional methods. 

The performed screening of MOFs demonstrated that Henry’s constants for He adsorption are in the range from 1.25 × 10^−8^ to 6.89 × 10^−6^ mol/kg/Pa, and for N_2_ adsorption they vary from 7.08 × 10^−8^ to 1.21 × 10^−3^ mol/kg/Pa. The N_2_/He adsorption selectivity at infinite dilution exceeds one for all the studied frameworks reaching a maximum value of 13,829. The highest Henry’s constants for N_2_ adsorption (Appendix A) and nitrogen adsorption selectivity (Appendix A) were observed at PLD about 4.5 Å and LCD about 6 Å. This result can be explained by the fact that the intermolecular interaction potentials overlap in small cavities leading to an increase in the heats of adsorption and Henry’s constants. The analysis of the relationship between structural parameters and Henry’s constants for He adsorption (Appendix A) revealed a trend to the $K_{He}^{0}$ growth with an increase in accessible surface area, porosity and accessible pore volume. Meanwhile, a trend to the $K_{He}^{0}$ decrease with an increase in the framework density was observed. In the case of $K_{N2}^{0},$ these trends were less evident due to the presence of an additional contribution from electrostatic interaction between the quadrupole moment of the nitrogen molecule and partial atomic charges of the MOF atoms. The analysis of the correlation between the structural parameters and N_2_ adsorption selectivity (Appendix A) showed a trend of the decrease in $S_{ads,N2/He}^{0}$ with an increase in accessible surface area and porosity, and a decrease in density. Similar relationships between the structural parameters and adsorption selectivity to nitrogen were also observed for zeolites (Appendix A). The highest adsorption selectivity to nitrogen calculated at infinite dilution for zeolites was as high as 26. A large variety of the MOFs chemical composition leads to a greater variety of their structural parameters compared to zeolites (Appendix A). As a result, the selectivity of the most promising MOFs in adsorption-based separation of nitrogen–helium mixtures exceeds the best of the zeolite frameworks by several orders of magnitude. 

To move from the estimation of ideal adsorption properties of MOFs and zeolites at infinite dilution to gas mixtures, adsorption of an equimolar mixture of helium and nitrogen was simulated at 0.01, 0.1, 0.3 and 1 Mpa and 298 K. The obtained results (Appendix A) demonstrate that at low adsorption selectivity $S_{ads,N_{2}/He}^{0}$ and $S_{ads,N_{2}/He}^{mix}$ are almost equal. However, when the adsorption selectivity increases, the $S_{ads,N_{2}/He}^{mix}$/$S_{ads,N_{2}/He}^{0}$ ratio becomes less than one. When the pressure increases, the difference between the two selectivity values grows as well. So, the higher the pressure in the system, the greater the effect of competitive adsorption. 

Two models of He/N_2_ adsorption-based separation at room temperature were considered in this study. In the first model, corresponding to the conditions of vacuum swing adsorption, the adsorption pressure was equal to 0.1 Mpa and the desorption pressure was equal to 0.01 Mpa. In the second model, corresponding to conditions of pressure swing adsorption, the adsorption pressure was equal to 1 Mpa and the desorption pressure was equal to 0.1 Mpa. As regenerability tended to decrease with the APS increase, only frameworks with regenerability above 80% were considered. The highest APS values were observed for frameworks with high nitrogen Δ*N* and $S_{ads,N_{2}/He}^{mix}$. Top 50 best MOFs have APS > 57 mol/kg and APS > 70 mol/kg for the VSA and PSA gas separation models, respectively. (Figure 2a,b). In the case of PSA, the APS metric of the most promising MOFs is higher by more than an order of magnitude than that of the most efficient zeolites (Figure 2d). In the case of VSA, this difference reaches two orders of magnitude (Figure 2c). This result clearly demonstrates how promising MOFs are for adsorption-based separation of helium and nitrogen. In addition to the APS metric, sometimes a more complex API (Adsorbent Performance Indicator) metric [45], which additionally takes into account enthalpy of adsorption, is used in the literature:(1)$$API=\frac{{\left(S_{ads,N_{2}/He}^{mix}-1\right)}^{A}\cdot \Delta N_{N2}{}^{B}}{{\left|\Delta H_{ads,N2}\right|}^{C}}$$

Here, $\Delta H_{ads,N2}$ is enthalpy of nitrogen adsorption; constants *A*, *B* and *C* are equal to 0.5, 2 and 1 [45]. As adsorption is an exothermal process, for large-scale industrial installations for separation based on vacuum swing adsorption, the use of an adsorbent with high adsorption enthalpy of results in an increase of the adsorber temperature leading to the decrease of the target component adsorption. On the other hand, heat is absorbed during the adsorbent regeneration leading to the decrease in the adsorber temperature, which makes the adsorbent regeneration more difficult. So, the higher the enthalpy of adsorption, the greater the difference of regenerability from the ideal model value. Therefore, an efficient adsorbent should have high adsorption capacity, high selectivity and low enthalpy of adsorption. In this study the API metric was calculated for all the studied MOFs and zeolites with regenerability above 80%. Appendix A demonstrates a linear correlation between API and APS for both VSA and PSA. So, the use of both metrics leads to the same set of the most promising frameworks.

In this study, structure–adsorption performance relationships were investigated. Two databases with Top 50 MOFs based on the APS metric were constructed for the VSA and PSA separation processes. Then, smoothed probability density distributions (PDF) were built for several structural parameters for the database of all studied MOFs and for the databases of Top 50 MOFs for the VSA and PSA separation processes. These structural parameters included pore limiting diameter, largest cavity diameter, accessible surface area, accessible pore volume, density and porosity. The range of effective structural parameters was determined using a criterion that this range included more than 90% of Top 50 MOFs. The importance of each structural parameter was estimated by comparing PDFs for all studied MOFs and Top 50 MOFs. If PDFs were about the same for both databases, this structural parameter was considered to have little effect on the adsorption performance. Meanwhile, if PDFs were very different and PDF for TOP 50 MOFs had a narrow distribution, it was possible to claim that certain range of optimal parameters existed. For instance, Figure 3 demonstrates that for the VSA separation process there is a narrow range 3.75 Å < PLD < 4.8 Å corresponding to more than 90% of Top 50 MOFs, and PDF for them is different from PDF for all MOFs. So, the following optimal structural parameters were determined for VSA (Appendix A): 3.75 Å < PLD < 4.8 Å, 4.4 Å < LCD < 6 Å, 100 m^2^/g < ASA < 700 m^2^/g, 0.02 cm^3^/g < AV < 0.09 cm^3^/g, 1300 kg/m^3^ < density < 2500 kg/m^3^, 0.04 < VF < 0.15. For PSA the optimal ranges are wider (Appendix A): 3.75 Å < PLD < 6.3 Å, 4.2 Å < LCD < 7.1 Å, 300 m^2^/g < ASA < 1400 m^2^/g, 0.03 cm^3^/g < AV < 0.18 cm^3^/g, 900 kg/m^3^ < density < 2500 kg/m^3^, 0.05 < VF < 0.2. To determine the chemical composition–adsorption performance relationships, the probabilities of finding certain metal atoms in the MOF structure were calculated for all the studied MOFs and the database of Top 50 MOFs for the VSA and PSA separation processes (Figure 4). Ga, Ru, V, Er, Gd, La, U and Ca were found to be the most suitable metals for the VSA separation model based on the probabilities of their presence among Top 50 MOFs and all studied MOFs. Meanwhile, the most widespread metal atoms Zn, Cu, Cd, Co and Mn (present in 54.7% of all MOFs) were scarcely present among Top 50 MOFs (less than 10%). In the case of the PSA model, U, Al, Er, Be and Mg were found to be the most suitable metals. Characteristics of the most promising MOFs (Top 10) for the VSA and PSA separation processes are reported in Appendix A. As the pressure increase leads to a much more significant selectivity decrease than the adsorption capacity increase, the best MOFs for VSA have higher APS than the most promising MOFs for PSA.

It is interesting to compare the results of the MOF screening obtained in this study with earlier literature data. A total of 213 MOFs were studied for VSA gas separation with very similar operational conditions (equimolar mixture of helium and nitrogen, gas pressure during the adsorption cycle 1.2 bar, pressure during the adsorbent regeneration 0.1 bar) [44]. UVEXAV was found to be the most promising MOF with API = 680 and adsorption selectivity $S_{ads,N_{2}/He}^{mix}$ = 222.7. In our study we discovered 24 MOFs with superior API metric and 34 MOFs with higher nitrogen selectivity. Additionally, in the same earlier study [44] it was demonstrated that the electrostatic contribution for most MOFs was negligible. Despite the fact that a nitrogen molecule has a relatively a low quadrupole moment, we believe that it is important to consider the electrostatic interactions between nitrogen molecules and the framework atoms during the screening. To estimate their contributions, Henry’s constants and nitrogen enthalpies of adsorption were calculated with the account of electrostatic interactions and without them (Appendix A). The account of electrostatic interactions results in the growth of the median value of Henry’s constants and nitrogen adsorption enthalpies by 10.5% and 3.4%, respectively. Due to the great variety of the MOFs chemical composition, larger spread in Henry’s constants and nitrogen adsorption enthalpies was observed for them in comparison with zeolites. In this study, it was found that the account of electrostatic interactions results in the increase of the Henry’s constants by at least a factor of 1.5 for 15% of all considered MOFs. Meanwhile, the analysis of Top 50 MOFs for VSA and PSA revealed a similar increase in Henry’s constants for 80% and 58% of MOFs, respectively. So, the contribution of electrostatic interactions between quadrupole moments of nitrogen molecules and the MOF atoms is significant for the most promising MOFs.

Since the chemical composition of both Top 50 MOFs includes a large number of different metal atoms with different atomic weights and partial charges, it can be assumed that metal atoms affect the adsorption performance primarily through the structure topology rather than through their contribution to the intermolecular interaction. Therefore, MOF screening for adsorption-based separation of N_2_/He should be based primarily on the search for MOFs with optimal structural parameters. Thus, an ideal adsorbent should have uniformly narrow pores without pockets or cavities and a significant density in order to provide high nitrogen adsorption enthalpy and N_2_/He selectivity. As a result, such an ideal structure will have low porosity, pore volume and accessible surface area.

### 2.2. Membrane-Based Gas Separation

To test the force fields used in this study for prediction of diffusion properties of MOFs and zeolites, calculated He and N_2_ permeances in the temperature range of 298–301 K were compared with the corresponding literature data [22,26,29,30,66,67,68,69,70,71]. Appendix A demonstrates that the models used in this study can be applied only for estimation of the membrane permeance. In addition to the lack of ideality in the used force field models, there are several additional reasons leading to deviation of the calculated permeance values from the experimental ones. First, most polycrystalline membranes based on MOFs and zeolites have low selectivity and high permeance due to the presence of defects both inside their framework and in the packing of crystals. The majority of membranes demonstrate selectivity close to that of the Knudsen diffusion model, which indicates that the size of pores between the crystallites can exceed 2 nm. Second, only permeability can be calculated by simulations. To connect this value with permeance, it is necessary to know the membrane thickness. Except for single-crystal membranes [27], the thickness of the MOF or zeolite layer on the support can be substantially varied resulting in deviation between the calculated and experimental values. Third, an ideal model imitating the adsorbate behavior at infinite dilution was used to calculate permeability in this study. Such a model does not take into account the effects of competitive adsorption and gas mixture composition on the diffusion properties. As helium is weakly adsorbed, and its concentration determined by GCMC is substantially lower than the N_2_ concentration, it is necessary to consider much larger framework fragments than for simulation of adsorption to obtain correct values of self-diffusion coefficients. In turn, this leads to a significant increase in the simulation time. Therefore, it is rational to use the ideal model at the first screening stage with the following correction of the membrane permeability and selectivity for Top 10 frameworks using Equations (10) and (11).

At the first stage of screening for the most promising materials for membrane-based separation of He and N_2_, diffusion coefficients were calculated at infinite dilution conditions. The analysis of the dependence of self-diffusion coefficients on the structural properties of the studied MOFs (Appendix A) demonstrates a natural trend towards an increase in self-diffusion coefficients with increasing pore size (PLD and LCD) and porosity. The greatest difference between $D_{N_{2}}^{0}$ and $D_{He}^{0}$ was observed in the PLD range under 6 Å and LCD range under 8 Å. The diffusion selectivity $S_{dif\;He/N_{2}}^{0}$ in such small pores can reach several orders of magnitude (Appendix A). This result can be explained by the difference in the kinetic diameters of helium and nitrogen molecules. Similar relationships between structural parameters and diffusion selectivity were also observed for zeolites (Appendix A). Similar to the adsorption-based separation, the screening demonstrated that great variety of MOFs makes it possible to find MOFs with much higher diffusion selectivity than that of zeolites (maximum $S_{dif\;He/N_{2}}^{0}$ for MOFs is equal to 1197 compared to 29 for zeolites).

The analysis of the screening results showed that the membrane selectivity $S_{mem\;He/N_{2}}^{0}$ varies from 2.4 × 10^−3^ to 4.6. Most of MOF membranes (79.9%) are N_2_ selective. The upper part of the MOF “cloud” in Figure 5 with the membrane selectivity $S_{mem\;He/N_{2}}^{0}\;$> 1 is characterized by predomination of the diffusion selectivity $S_{dif\;He/N_{2}}^{0}$ over the adsorption selectivity $S_{ads\;He/N_{2}}^{0}$. The reverse ratio characterizes the bottom part of the MOF “cloud” in Figure 5. Helium permeability $P_{He}^{0}$ is in the range from 2864 to 2.9 × 10^6^ Barrer, whereas $P_{N_{2}}^{0}$ is in the range of 4034–1.4 × 10^7^ Barrer. The highest permeabilities are observed for wide-pore frameworks. For examples, in Figure 5 one can see a “tail” consisting of 50 MOFs with $P_{He}^{0}$ > 5 × 10^5^ Barrer. MOFs in this “tail” have PLD from 6.9 to 70.8 Å and LCD from 7 to 70.9 Å. Diffusion in these MOFs either follows the Knudsen diffusion model or is close to it. The diffusion selectivity in this diffusion mode is determined as $S_{dif,i/j}^{\mathrm{Knudsen}}=\sqrt{M_{j}/M_{i}}$. For separation of He and N_2_, this value is equal to 2.65.

In addition, the effect of electrostatic interaction on self-diffusion coefficients and nitrogen permeability as well as membrane selectivity to helium was studied (Appendix A). The account of electrostatic interaction results in a minor decrease in $D_{N_{2}\;\left(charged\right)\;}^{0}$ relative to $D_{N_{2}\;\left(non-charged\right)\;}^{0}$. As electrostatic interaction has a more significant effect on Henry’s adsorption constants compared to self-diffusion coefficients, on the average the nitrogen permeability grows after the account of electrostatic interaction. As a result, the membrane selectivity to helium decreases. So, electrostatic interactions between the quadrupole moment of nitrogen molecules and the framework atoms should be taken into account during both adsorption-based and membrane-based screening.

The perspectives of using MOFs and zeolites as membrane materials for separation of He and N_2_ can be estimated by comparing calculated selectivity and permeability with the upper bonds obtained in different years for polymeric membranes. Upper bond is an empirical relation $P_{i}=kS_{mem\;i/j}{}^{n}$ where constants *k* and *n* are determined from the selectivity vs. permeability graph for experimental data on He/N_2_ separation over various polymeric membranes. Figure 6 presents upper bonds obtained by Robeson in 1991 [72] and 2008 [12] and by Wu et al. in 2019 [73]. The upper bonds for polymeric membranes demonstrate that the best MOFs and zeolites trail polymeric membranes in selectivity $S_{mem\;He/N_{2}}^{0}$ but have higher permeability $P_{He}^{0}$. Based on this fact, several variants of using MOFs and zeolites for separation of He and N_2_ are possible: creation of materials from a combination of polymers and MOFs/zeolites or using MOF/zeolite materials selective to nitrogen. In the former case, combination of materials can result in the growth in permeability (in comparison with polymeric membranes) and increase in helium selectivity (in comparison with MOFs/zeolites). In several publications [49,50,51] it was shown that such an approach makes it possible to obtain more efficient materials for separation of different gas mixtures than using MOFs along as the membrane materials. However, due to the low membrane selectivity of MOFs and zeolites to helium, apparently, any other mesoporous materials with the membrane selectivity close to the Knudsen diffusion selectivity can be used for separation of helium and nitrogen. In the second case, if MOFs or zeolites selective to nitrogen are used as the membrane, the retentate will be enriched with helium and the permeate will be enriched with nitrogen. As using MOFs and zeolites as membrane materials is more promising in the second case, Top 50 frameworks most selective to nitrogen ($S_{mem\;N_{2}/He}^{0}$ > 22.6) were identified. Not a single zeolite was included in this list.

Structure–membrane separation performance relationships were also studied. Their analysis was performed using the same technique that was used for adsorption-based separation and was described above in Section 3.1. The following optimal structural parameters were determined for membrane-based separation (Appendix A): 3.75 Å < PLD < 4.65 Å, 4 Å < LCD < 5.8 Å, 100 m^2^/g < ASA < 450 m^2^/g, 0.01 cm^3^/g < AV < 0.06 cm^3^/g, 1100 kg/m^3^ < density < 2400 kg/m^3^, 0.02 < VF < 0.1. Note that the PDF functions for Top 50 MOFs and all the studied MOFs are substantially different, and the range of optimal structural parameters is much narrower than for adsorption-based separation. Similarly, chemical composition–membrane separation performance relationships were analyzed. Top 50 MOFs include 22 metals. It means that the membrane properties are primarily determined by its structural parameters rather than by chemical composition. Still, based on the ratio of probabilities of the presence among Top 50 MOFs and in all the studied MOFs, Ga and Al were found to be the most promising metals for membrane-based separation.

The characteristics of the most promising MOFs (Top 10) for membrane-based separation are reported in Appendix A. All MOFs from Top 10 demonstrate high adsorption selectivity predominating over their diffusion selectivity and high N_2_ permeability exceeding the He permeability by more than an order of magnitude. At the second stage of screening, these MOFs were used in simulation by GCMC and MD using an equimolar mixture of He and N_2_ at 298 K and 3 bar, which corresponds to the conditions of real membrane-based separation of gases. The simulation results (Appendix A) indicate that the switch to the equimolar mixture leads to simultaneous decrease of the nitrogen adsorption selectivity, He/N_2_ diffusion selectivity and permeability. Nevertheless, the values of $S_{mem\;N_{2}/He}^{0}$ and $S_{mem\;N_{2}/He}^{mix}$ are very similar indicating that the suggested technique used for the initial screening makes it possible to predict promising frameworks while using minimum computer power. Earlier [44], based on the results of in silico screening of 500 MOFs, the highest membrane selectivity $S_{mem\;N_{2}/He}^{0}$ = 44.91 was found for LIFWOO. In this study, we determined 16 MOFs with even higher membrane selectivity. Thus, highly nitrogen-selective membranes are characterized by the predominance of the N_2_/He adsorption selectivity over N_2_/He diffusion selectivity. Therefore, as in the case of adsorption-based separation, an ideal membrane should have narrow uniform pores and a significant density. The membrane properties are primarily determined by its structural parameters rather than by chemical composition.

## 3. Computational Methods

### 3.1. MOF and Zeolite Databases

“Computation Ready, Experimental Metal−Organic Framework Database” (CoRE MOF 2019) [34] was used as the parent MOF database. At the first step only ordered frameworks where all solvent molecules were removed were selected from this database. The resulting database consisted of 10143 MOFs. Then, the Zeo++ algorithm [74] was used to calculate density, porosity (VF), accessible surface area (ASA), pore limiting diameter (PLD) and largest cavity diameter (LCD) for each framework. Additionally, Zeo++ was used to identify MOFs with pockets not accessible both for helium and nitrogen molecules. In the following computations, such pockets were blocked. The accessible pore volume (AV) was determined using RASPA-2.0 software package [75] with helium molecule used as a probe. Further, only MOFs with non-zero ASA and PLD > 3.75 Å were selected from the database. These conditions would allow both gases to penetrate through the membrane (the kinetic diameter is equal 3.64 Å for N_2_ and 2.56 Å for He). So, the number of analyzed MOFs was shortened to 5156. 

IZA database [32] implemented in iRASPA visualization package [76] was used as the parent database of all-silica zeolites and zeolite-like materials. Structural parameters were determined for all frameworks from this database using a procedure similar to the one used earlier for MOFs. After removing frameworks with PLD below 3.75 Å, the number of analyzed zeolite frameworks was reduced to 110. 

### 3.2. Simulation Details

The adsorption and diffusion properties of the studied frameworks were simulated using equilibrium molecular dynamics (EMD) and Grand canonical Monte Carlo (GCMC) methods in the RASPA-2.0 package. The interactions were described by the sum of Lennard-Jones 6–12 (LJ) and Coulomb interaction potentials. The nitrogen molecule was simulated according to the TraPPE [77] force field as a dumbbell with a rigid bond between the atoms with the length of 1.1 Å. The LJ interaction parameters for each atom were *ε_N_/k_B_* = 36.0 K and *σ_N_* = 3.31 Å. The N_2_ quadrupole moment was described by three charges: two *–q* charges where *q* = 0.482e that were located at the centers of the nitrogen atoms and +2*q* charge located at the molecule center of masses. The helium molecule was simulated by a single-charge model [78] with parameters *σ_He_* = 2.64 Å and *ε_He_/k_B_* = 10.9 K. The constants of the LJ 6–12 potential for atoms in MOFs were simulated according to the Dreiding [60] force field. If the necessary parameters were missing, the required data were retrieved from the UFF [79] force field. The force field developed by Vujić and Lyubartsev [62] was used for simulation of adsorption and diffusion in zeolites. Cross constants of the LJ interaction were calculated using the Lorentz-Berthelot mixing rule. The LJ interaction was cut at the distance *R_cutoff_* = 12.8 Å. Its potential was shifted to zero starting from the distance 0.9 *R_cutoff_*. Determination of partial atomic charges by REPEAT [80] or DDEC [81] methods that have high precision requires periodic DFT calculations for each studied structure. A detailed and systematic analysis of the effect of the choice of framework partial atomic charges on CO_2_ adsorption in six different widely studied MOFs predicted by molecular simulations was performed in the study [82]. It was shown that the partial charges obtained by both DDEC and REPEAT methods yielded isotherms that were in good agreement with each other. In our previous studies, it was shown that partial charges obtained by DDEC and REPEAT methods lead to similar results for modeling hydrogen adsorption in SAPO-11 (zeolite-like material) at 77 K [83] and water adsorption in CAU-10-H (MOF) at 298 K [84]. Thus, it can be expected that the use of the DDEC or REPEAT methods for calculating atomic partial charges will lead to similar results in the case of modeling nitrogen adsorption in MOFs and zeolites. A reasonable alternative is to use pre-trained models obtained by machine learning based on the CoRE MOF DDEC [43,85] containing 2932 optimized structures with partial atomic charges calculated by the DDEC method. So, partial atomic charges of the framework atoms were determined using pre-trained Random Forest model in the PACMOF code [86]. Long-range Coulomb interactions were calculated using the Ewald summation technique. The size of the MOF and zeolite structure fragments was selected to ensure the minimum distance in each direction exceeded 2*R_cutoff_*. In each simulation, we assumed that the MOF or zeolite structure was rigid and did not contain any defects.

Henry’s constants were calculated using the Widom particle insertion method. Henry’s constants were calculated at infinite dilution conditions for 10^5^ cycles at 298 K. The adsorption selectivity at infinite dilution $S_{ads,i/j}^{0}$ was determined as the ratio of Henry’s constants $K_{i}^{0}$ [10]:(2)$$S_{ads,i/j}^{0}=K_{i}^{0}/K_{j}^{0}$$

The adsorption selectivity for separation of gas mixtures $S_{ads,i/j}^{mix}$ was calculated for 0.01, 0.1, 0.3 and 1 MPa pressures at 298 K as follows [10]:(3)$$S_{ads,i/j}^{mix}=\frac{N_{i}/N_{j}}{x_{i}/x_{j}}$$

Here, *N_i_* is the adsorption of the *i*-th component determined from GCMC, *x_i_* is the molar fraction of the *i*-th component. For a gas mixture, the adsorption values *N_i_* were simulated using the following GCMC moves: translation, rotation, insertion, deletion, reinsertion, identity exchange of He and N_2_ molecules. The fugacity coefficient was calculated from the Peng–Robinson equation of state. 

Adsorbent Performance Score (APS) used to estimate the adsorbent efficiency was calculated as follows [87]:(4)$$APS=\Delta N_{N2}S_{ads,N2/He}^{mix}$$

The sorbent regenerability was calculated as [10]:(5)$$R=\Delta N_{N2}/N_{ads,N2}$$

Self-diffusion coefficient was determined by equilibrium molecular dynamics from the root-mean-square particle displacement using the following formula [88]:(6)$$D_{self,i}=\underset{t\to \infty }{\lim}\frac{1}{2dt}\langle \frac{1}{N}\overset{N}{\underset{j=1}{\sum }}{\left(\vec{r_{j}}\left(t\right)-\vec{r_{j}}\left(0\right)\right)}^{2}\rangle$$

Here, *N* is the number of molecules, $\vec{r_{j}}\left(t\right)$ is the position of the *j*-th particle at the moment *t*, *d* is the dimension of the system. Self-diffusion coefficient at infinite dilution $D_{i}^{0}$ was simulated by positioning 30 adsorbate molecules in the MOF fragment with all interactions between the gas molecules switched off. The simulation was performed in the NVT ensemble (at constant number of particles, volume and temperature) using the Nosé–Hoover thermostat. These conditions simulate the properties of the adsorbate at infinite dilution. After the initial position of the adsorbate in the framework was generated, for the first 50 ps the system was subjected to equilibration before the data collection. The MD simulation time was 20 ns. The diffusion selectivity at infinite dilution $S_{dif,i/j}^{0}$ was calculated as follows [42]:(7)$$S_{dif,i/j}^{0}=D_{i}^{0}/D_{j}^{0}$$

Permeability $P_{i}^{0}$ was determined using Equation (7) [42]:(8)$$P_{i}^{0}=D_{i}^{0}K_{i}^{0}$$

Membrane selectivity $S_{mem,i/j}^{0}$ was estimated as follows [42]:(9)$$S_{mem,i/j}^{0}=P_{i}^{0}/P_{j}^{0}$$

For selected Top 10 frameworks, self-diffusion coefficient *D_self,i_* was calculated for each component of the gas mixture at the adsorbate concentrations determined by GCMC. As helium is weakly adsorbed, and its concentration determined by GCMC is significantly lower than that of N_2_, the framework fragment used in the EMD method was 27 times larger than during the initial screening at infinite dilution. Five independent EMD runs were performed to collect statistics. Diffusion selectivity for a mixture of gases $S_{dif,i/j}^{mix}$ was determined as the ratio of self-diffusion coefficients [42]:(10)$$S_{dif,i/j}^{mix}=\frac{D_{self,i}}{D_{self,j}}$$

Permeability of the *i*-th mixture component $P_{i}^{mix}$ was calculated according to Equation (10) [42]:(11)$$P_{i}^{mix}=\frac{\varphi \cdot D_{self,i}\cdot c_{i}}{f_{i}}$$

Here, *φ* is the adsorbent porosity, *c_i_* is the concentration of the *i*-th mixture component at the feed side of the membrane determined by GCMC, *f_i_* is the fugacity of the *i*-th mixture component before the membrane. This model assumes vacuum permeating pressure. The membrane selectivity for the mixture of gases $S_{mem,i/j}^{mix}$ was determined as follows [42]:(12)$$S_{mem,i/j}^{mix}=P_{i}^{mix}/P_{j}^{mix}$$

## 4. Conclusions

In silico screening of 10143 MOFs and 218 all-silica zeolites for adsorption-based and membrane-based separation of He and N_2_ was performed. GCMC and EMD methods were used to calculate Henry’s constants, adsorption at different pressures and self-diffusion coefficients for He and N_2_. These values were used to calculate major metrics, such as adsorption, diffusion and membrane selectivity, regenerability and permeability. Dependence of adsorption selectivity $S_{ads,N_{2}/He}^{mix}$ in an equimolar mixture of He and N_2_ on the pressure in the system was studied. The effect of electrostatic interaction between the quadrupole moment of nitrogen molecules and framework atoms on the main adsorption and diffusion metrics was studied. MOFs were demonstrated to be more efficient than zeolites in both adsorption-based and membrane-based separation. Top 10 most promising MOFs for membrane-based, VSA and PSA separation methods were determined. The analysis of structure–adsorption and membrane performance relationships made it possible to determine the ranges of structural parameters, such as pore limiting diameter, largest cavity diameter, surface area, porosity, accessible surface area and pore volume, corresponding to the most promising MOFs for each separation model considered in this study. A similar analysis was performed to find out the optimal MOF chemical composition. The screening performed in this study can be called topological because the structural flexibility, possible presence of structural defects or modification of linkers with functional groups were not taken into account. Similarly, in the case of zeolites possible changes of the Si/Al ratio, variation of the cation composition or modification of their structure by isomorphous substitutions were not considered. Despite these limitations, one can expect that the ranges of optimal structural parameters and trends of adsorption and membranes metrics determined in this study will be correct even if all these factors are taken into account. The obtained results can be used as a guide for selection of frameworks for He/N_2_ separation.

## Figures and Tables

**Figure 1 molecules-28-00020-f001:**
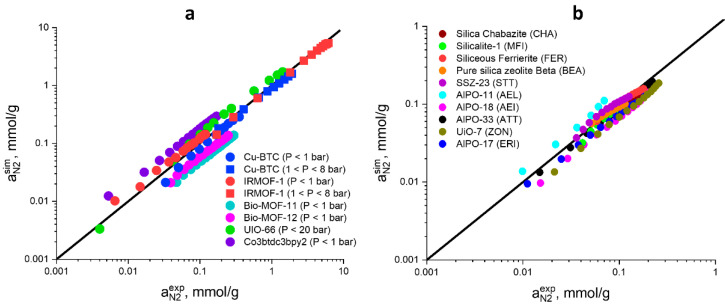
Comparison of experimental and molecular simulation data for N_2_ adsorption in MOFs (**a**) and zeolites (**b**).

**Figure 2 molecules-28-00020-f002:**
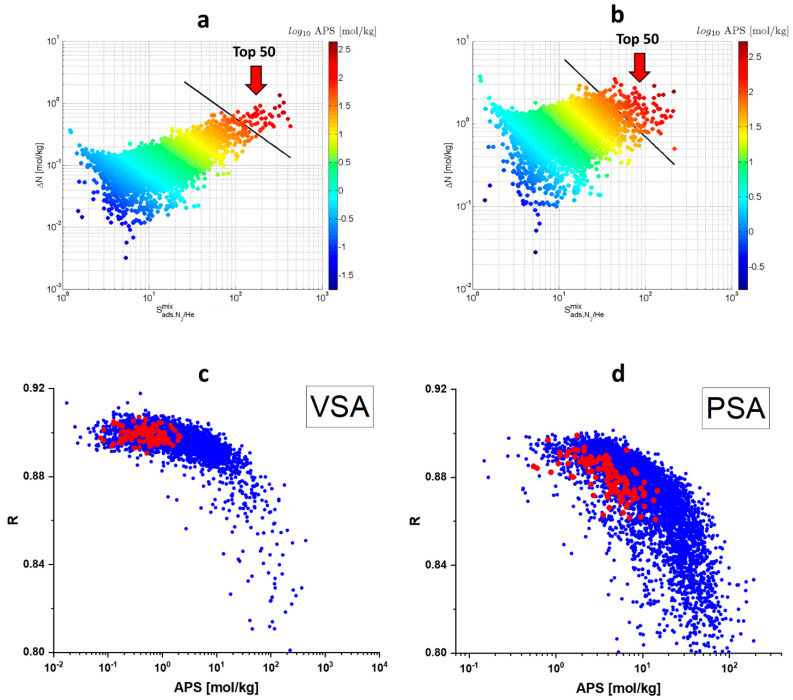
Dependence of the working capacity (ΔN) on adsorption selectivity ($S_{ads,N_{2}/He}^{mix}$) and APS in the case of vacuum swing adsorption (**a**) and pressure swing adsorption (**b**). Dependence of the regenerability (*R*) on APS for MOFs (blue) and zeolites (red) in the case of vacuum swing adsorption (**c**) and pressure swing adsorption (**d**).

**Figure 3 molecules-28-00020-f003:**
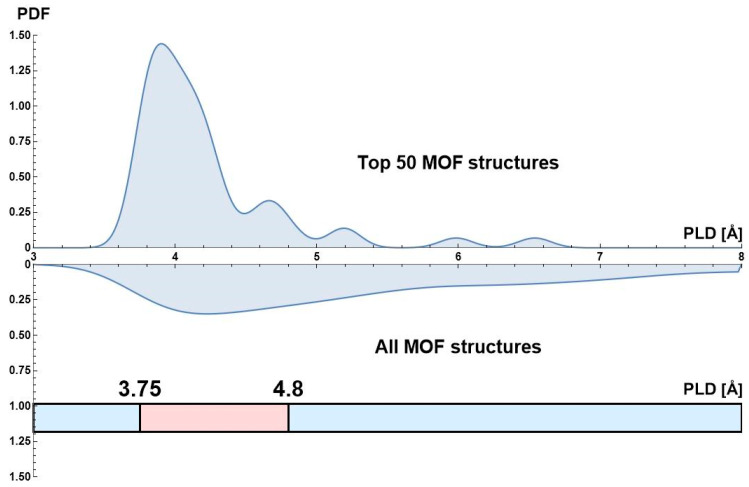
Comparison of the smoothed probability density distribution (PDF) of pore limiting diameter (PLD) for the best 50 MOFs and for all MOFs considered in this study for the VSA separation process. At the bottom of the figure is the PLD range that comprises over 90% of the Top 50 MOFs database.

**Figure 4 molecules-28-00020-f004:**
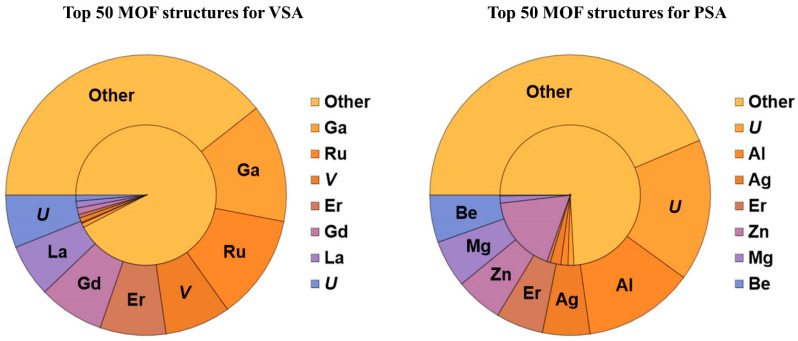
Chemical composition distribution of MOF structures by metal type. The outer pie chart corresponds to the Top 50 best MOFs, the inner one corresponds to all MOFs considered in this study.

**Figure 5 molecules-28-00020-f005:**
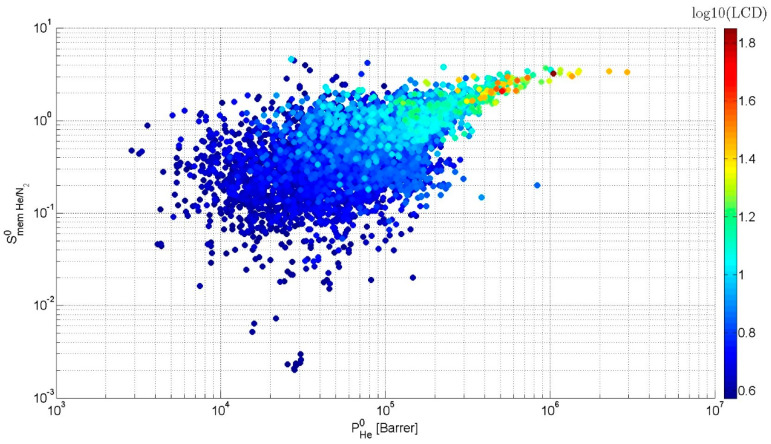
Dependence of membrane selectivity ($S_{mem\;He/N_{2}}^{0}$) on helium permeability ($P_{He}^{0}$) determined at infinite dilution and PLD.

**Figure 6 molecules-28-00020-f006:**
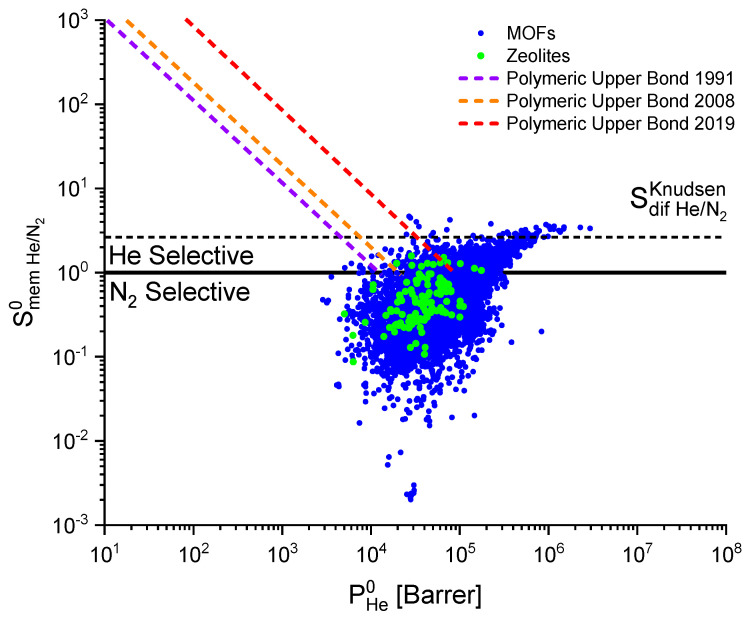
Dependence of membrane selectivity ($S_{mem\;He/N_{2}}^{0}$) on helium permeability ($P_{He}^{0}$) determined at infinite dilution. Lines correspond to the upper bonds for polymeric materials.

## Data Availability

Data available on request.

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
