# Peer review of "In Silico Screening of Metal−Organic Frameworks and Zeolites for He/N2 Separation"

_molecules, 2022, doi:10.3390/molecules28010020_

Round 1
Reviewer 1 Report
Ivan V. Grenev et al. have performed of 10143 MOFs and 218 all-silica zeolites for adsorption-based and membrane-based separation of He and N2. Moreover, GCMC and EMD methods were used to calculate Henry's constants, adsorption at different pressures and self-diffusion coefficients for He and N2. It is novel and interest to the researchers in the related areas. I would consider the paper for publication after minor revisions are made according to the following specific comments:
1. In Fig. 1, the words on the Y-axis are messy.
2. The mechanism of He/N2 separation should be further discussed.
3. The corresponding formula should be marked with references.
4. For the study of separation, the authors may refer these papers: 1. Nanoscale, 2017, 9, 14229; 2. Nanoscale 11 (38), 17607, 2019
5. For more perfection, several language mistakes could be revised.
Author Response
Dear Assistant Editor Hoshi Wang
Thank you for your letter and for the valuable comments of the Reviewers concerning our manuscript. We have done our best to address all the reviewers’ comments and revise the manuscript accordingly. I am sending you our responses on the comments of the reviewers on the manuscript text. All corrections were made by us in the updated version of the article. Below are our point-to-point answers for all the comments.
Reviewer # 1:
- In Fig. 1, the words on the Y-axis are messy.
We revised Figure 1, 2 and 4.
- The mechanism of He/N2 separation should be further discussed.
The discussion of the relationship of separation mechanisms between the structural characteristics and the chemical composition of MOFs have been introduced into the text (аt the end of sections 3.1 and 3.2).
- The corresponding formula should be marked with references.
The references have been added for all formulas used in the work.
- For the study of separation, the authors may refer these papers: 1. Nanoscale, 2017, 9, 14229; 2. Nanoscale 11 (38), 17607, 2019.
We carefully read the articles recommended by the reviewer and considered it possible to mention second study (Nanoscale 11 (38), 17607, 2019) in our manuscript.
- For more perfection, several language mistakes could be revised.
The authors carefully checked the text of the manuscript and made some corrections.

Reviewer 2 Report
In this manuscript, the authors screened large number of MOFs and zeolites for adsorption- and membraned-based He and N2 separation by combining Grand Canonical Monte Carlo (GCMC) with Equilibrium Molecular Dynamics (EMD), and investigated the structure-performance relations. Through detailed analysis, the authors determined top 10 most promising MOFs for different separation methods, and studied the electrostatic interaction between N2 molecules and MOF atoms at last. Herein, I would like to discuss the following points with the authors, and this work can be considered for publication when the following questions are solved.
1. The metrics of non-zero ASA and PLD was selected for the prescreening of MOFs and zeolites, where the value of PLD (3.75 Å) is larger than the kinetic diameter of N2 (3.64 Å) and He (2.56 Å), but is it feasible to select a slightly smaller or larger one such as 3.70 Å or 3.80 Å or larger?
2. In the section of computational methods, partial atomic charges of framework atoms were considered by DDEC methods, and whether the reliability of this method is verified? Or the REPEAT method was applied for comparison?
3. Besides, there are also some other problems: (1) These two keywords “metal-organic frameworks” and “MOFs” are repeated; (2) The formatting of the article title in Ref. 16 is inconsistent with that in the other references.
Author Response
List of responses
Dear Assistant Editor Hoshi Wang
Thank you for your letter and for the valuable comments of the Reviewers concerning our manuscript. We have done our best to address all the reviewers’ comments and revise the manuscript accordingly. I am sending you our responses on the comments of the reviewers on the manuscript text. All corrections were made by us in the updated version of the article. Below are our point-to-point answers for all the comments.
Reviewer #2:
- The metrics of non-zero ASA and PLD was selected for the prescreening of MOFs and zeolites, where the value of PLD (3.75 Å) is larger than the kinetic diameter of N2 (3.64 Å) and He (2.56 Å), but is it feasible to select a slightly smaller or larger one such as 3.70 Å or 3.80 Å or larger?
If too small values of PLD parameter are used for the prescreening procedure, structures with a low nitrogen self-diffusion coefficient will be included in the initial database. Such structures will be ineffective for both adsorption-based and membrane-based separation. If we set too large values of PLD parameter, then some of the effective structures will be overlooked. The correctness of the choice of the value of PLD parameter can be assessed based on the results of screening. It was shown that the main probability density distribution peaks of PLD for the best 50 MOF structures for VSA, PSA, and membrane-based separation are located at PLD equal to 3.9 Å, 4.2 Å, and 3.95 Å, respectively. This indicates the correctness of the choice of the value of the PLD parameter.
- In the section of computational methods, partial atomic charges of framework atoms were considered by DDEC methods, and whether the reliability of this method is verified? Or the REPEAT method was applied for comparison?
In the study [1] was performed a detailed and systematic analysis of the effect of the choice of framework partial atomic charges on CO2 adsorption in 6 different widely studied MOFs predicted by molecular simulations. It was shown that the partial charges obtained by both DDEC and REPEAT methods yielded isotherms that were in good consistent with each other. In our previous studies, it was shown that partial charges obtained by DDEC and REPEAT methods lead to similar results for modeling hydrogen adsorption in SAPO-11 (zeolite-like material) at 77 K [2] and water adsorption in CAU-10-H (MOF) at 298 K [3]. Thus, it can be expected that the use of the DDEС or REPEAT methods for calculating atomic partial charges will lead to similar results in the case of modeling nitrogen adsorption in MOFs and zeolites. The relevant comment has been introduced into the text (in section 2.2).
References:
- Sladekova, K.; Campbell, C.; Grant, C.; Fletcher, A.J.; Gomes, J.R.B.; Jorge, M. The Effect of Atomic Point Charges on Adsorption Isotherms of CO2 and Water in Metal Organic Frameworks. Adsorption 2019, doi:10.1007/s10450-019-00187-2.
- Grenev, I.V.; Klimkin, N.D.; Shamanaeva, I.A.; Shubin, A.A.; Chetyrin, I.A.; Gavrilov, V.Y. A Novel Adsorption-Based Method for Revealing the Si Distribution in SAPO Molecular Sieves: The Case of SAPO-11. Microporous Mesoporous Mater. 2021, 328, 111503, doi:10.1016/j.micromeso.2021.111503.
- Grenev, I.V.; Shubin, A.A.; Solovyeva, M.V.; Gordeeva, L.G. The Impact of Framework Flexibility and Defects on the Water Adsorption in CAU-10-H. Phys. Chem. Chem. Phys. 2021, 23, 21329–21337, doi:10.1039/D1CP03242A.
3.Besides, there are also some other problems: (1) These two keywords “metal-organic frameworks” and “MOFs” are repeated; (2) The formatting of the article title in Ref. 16 is inconsistent with that in the other references.
We left only the "metal−organic frameworks" in the keywords section. We have corrected the formatting of the article title in Ref. 16 (after revision Ref. 19).

Round 2
Reviewer 1 Report
The authors addresssed the most issues well, and it could be accepted after the following minor comments are solved:1. several language mistakes could be revised. 2. For the study of separation, the authors may refer: Nanoscale, 2017, 9, 14229;
Author Response
Reviewer # 1:
1. Several language mistakes could be revised.
The authors carefully checked the text of the manuscript and made some additional corrections.
2. For the study of separation, the authors may refer: Nanoscale, 2017, 9, 14229.
We mentioned suggested study in our manuscript.
Reviewer 2 Report
The authors had revised or explained all the questions from the reviewers in the revised manuscript. So I recommend publication this paper in Molecules.
Author Response
Thank you for the valuable comments concerning our manuscript.